Pan-cancer systematic identification of lncRNAs associated with cancer prognosis

Ung Matthew 1
Schaafsma Evelien 1
Mattox Daniel 2
Wang George L. 1
http://orcid.org/0000-0002-5002-3417 Cheng Chao 1 3 4 5 chao.cheng@dartmouth.edu
1 Department of Molecular and Systems Biology, Dartmouth College , Hanover, NH , USA
2 Department of Computer Science, Dartmouth College , Hanover, NH , USA
3 Department of Medicine, Baylor College of Medicine , Houston, TX , USA
4 The Institute for Clinical and Translational Research, Baylor College of Medicine , Houston, TX , USA
5 Department of Biomedical Data Science, Geisel School of Medicine at Dartmouth , Lebanon, NH , USA
Piccolo Stephen
Electronic publication date: 2020 Mar 24
Publication date: 2020
Volume: 8
Electronic Location ID: e8797
Received 2019 Nov 21; Accepted 2020 Feb 25
Copyright: © 2020 Ung et al.
Copyright year: 2020
Copyright holder: Ung et al.
License: This is an open access article distributed under the terms of the Creative Commons Attribution License, which permits unrestricted use, distribution, reproduction and adaptation in any medium and for any purpose provided that it is properly attributed. For attribution, the original author(s), title, publication source (PeerJ) and either DOI or URL of the article must be cited.
License URL: https://creativecommons.org/licenses/by/4.0/

Keywords: LncRNA, Prognosis, Microarray, RNA-seq, TCGA

Funding: American Cancer Society Research #IRG-82-003-30 National Center for Advancing Translational Sciences of the National Institutes of Health KL2TR001088 Rosaline Borison Memorial Pre-Doctoral Fellowship Cancer Prevention Research Institute of Texas (CPRIT) RR180061 National Cancer Institute of the National Institutes of Health 1R21CA227996 CPRIT Scholar in Cancer Research This work was supported by American Cancer Society Research grant #IRG-82-003-30, the National Center for Advancing Translational Sciences of the National Institutes of Health under Award Number KL2TR001088, the Rosaline Borison Memorial Pre-doctoral fellowship provided to Matthew H. Ung, the Cancer Prevention Research Institute of Texas (CPRIT) (RR180061 to Chao Cheng) and the National Cancer Institute of the National Institutes of Health (1R21CA227996 to Chao Cheng). Chao Cheng is a CPRIT Scholar in Cancer Research. The funders had no role in study design, data collection and analysis, decision to publish, or preparation of the manuscript.

==============================
Background

The “dark matter” of the genome harbors several non-coding RNA species including Long non-coding RNAs (lncRNAs), which have been implicated in neoplasia but remain understudied. RNA-seq has provided deep insights into the nature of lncRNAs in cancer but current RNA-seq data are rarely accompanied by longitudinal patient survival information. In contrast, a plethora of microarray studies have collected these clinical metadata that can be leveraged to identify novel associations between gene expression and clinical phenotypes.

Methods

In this study, we developed an analysis framework that computationally integrates RNA-seq and microarray data to systematically screen 9,463 lncRNAs for association with mortality risk across 20 cancer types.

Results

In total, we identified a comprehensive list of associations between lncRNAs and patient survival and demonstrate that these prognostic lncRNAs are under selective pressure and may be functional. Our results provide valuable insights that facilitate further exploration of lncRNAs and their potential as cancer biomarkers and drug targets.

Introduction

Long non-coding RNAs (lncRNAs) constitute a relatively unexplored repertoire of gene products that exhibit diverse functions and are involved in several biological processes. As such, the ENCODE consortium reported that 80% of the genome is transcribed into a variety of functional products including non-coding RNAs (The ENCODE Project Consortium, 2012). Several high-level characteristics of lncRNAs provide evidence that they are indeed functional, including their association with chromatin signatures of active transcription, being transcribed by RNA polymerase II, and undergoing post-transcriptional modifications such as polyadenylation and alternative splicing (Wang & Chang, 2011; Rinn & Chang, 2012; Kung, Colognori & Lee, 2013). The mechanisms by which lncRNAs regulate biological processes have not been studied in detail but evidence suggest that they can function at the transcriptional, post-transcriptional and post-translational level by acting as biological signals, decoys, guides and scaffolds (Wang & Chang, 2011; Rinn & Chang, 2012; Kung, Colognori & Lee, 2013). Moreover, the organization of lncRNAs across the genome is quite diverse in that they can be transcribed from intergenic regions, sites anti-sense to protein coding genes, bi-directional promoters, or within gene introns (Ponting, Oliver & Reik, 2009; Kung, Colognori & Lee, 2013).

Having been previously referred to as transcriptomic noise or “junk” DNA, lncRNAs are now being investigated as molecular players in several disease processes including cancer (Mattick & Makunin, 2006; Esteller, 2011; The ENCODE Project Consortium, 2012; Sahu, Singhal & Chinnaiyan, 2015; Schmitt & Chang, 2016; Bartonicek, Maag & Dinger, 2016; Evans, Feng & Chinnaiyan, 2016). In this particular context, lncRNAs have been implicated in all hallmarks of cancer including sustaining proliferative signaling, evading growth suppressors, enabling replicative immortality, activating invasion and metastasis, inducing angiogenesis and resisting cell death (Hanahan & Weinberg, 2000; Gutschner & Diederichs, 2012; Ali et al., 2018; Chiu et al., 2018). Aberrant expression of lncRNAs might be due to their close association with certain key driver genes (Ashouri et al., 2016) or the establishment of cancer-specific genomic features in lncRNA loci itself, including mutational events, methylation, copy number and SNP events (Iyer et al., 2015; Yan et al., 2015). Several studies have performed pan-cancer screens for lncRNAs involved in the disease and found that several of them were differentially expressed compared to normal samples, revealing their potential as biomarker candidates (Cabanski et al., 2015; Yan et al., 2015; Byron et al., 2016; Ching et al., 2016). For instance, PCA3 is a lncRNA that is currently approved for clinical use as a prostate cancer diagnostic biomarker and can be detected in patient urine samples (De Kok et al., 2002). Thus, dissecting the molecular characteristics of these understudied RNAs and their associations with disease phenotypes may yield findings that can be translated into the clinic.

In light of these findings, there is a paucity of patient samples with matched RNA-seq data and clinical information which limits the ability to perform pan-cancer screening for prognostic lncRNAs. Furthermore, few of these matched datasets contains sufficiently long follow-up times which limits statistical power when performing survival analyses, especially in cancer types where patients exhibit high survival rates (Clark et al., 2003). In stark contrast, there is a plethora of microarray gene expression data that are available, many of which are accompanied by comprehensive clinical information with long follow-up times.

Thus, using primarily protein-coding gene expression from microarray to infer the expression of their non-coding counterparts can re-purpose these valuable data and generate novel hypotheses about lncRNAs associated with patient mortality across several cancer types. To this end, multiple studies have attempted to utilize data from microarray to make inferences about lncRNA activity and their clinical relevance. Du et al. (2013) re-annotated probes from microarray data to identify prognostic transcriptional activity for ~10,000 lncRNAs in prostate cancer, glioblastoma, ovarian cancer and lung squamous cell carcinoma. From this screen, they identified novel lncRNA markers of mortality risk and validated several of them experimentally. Furthermore, Guo, Yao & Jiang (2016) performed a “guilt-by-association” analysis whereby lncRNAs that share an edge with prognostic protein coding genes in a biological network defined a priori were also considered prognostic. Although these studies have provided valuable insights into lncRNA biology, the reannotation of microarray probes might have missed prognostic lncRNAs not captured by microarray probes. In addition, lncRNA inference based on known protein coding target genes might bias lncRNA expression if not all target genes are known.

Therefore, we introduce a lncRNA inference approach that generates cancer-specific weighted lncRNA regulon network profiles de novo using RNA-seq data from The Cancer Genome Atlas (TCGA), and apply them to infer lncRNA expression in the PRECOG (Gentles et al., 2015) and METABRIC (Curtis et al., 2012) microarray compendia, which provide expression of protein-coding genes but not for most lncRNAs. Afterwards, we systematically interrogated each lncRNA to identify those that significantly associate with patient prognosis using clinical metadata included in the microarray studies. In total we screened 9,463 unique lncRNAs across 20 different cancer types to identify novel associations.

Materials and Methods

Data source and pre-processing

The lncRNA gene list with Ensembl IDs was derived from the TANRIC resource (Li et al., 2015). Level 3 RNA-seq count data from tumor samples encompassing 23 different cancer types along with corresponding clinical information were downloaded from the National Cancer Institute’s Genomic Data Commons data portal (https://portal.gdc.cancer.gov/). The count data was normalized by library size and subjected to a variance stabilizing transformation implemented using DESeq2 (Love, Huber & Anders, 2014). This transforms the expression values so that they are homoskedastic by fitting the dispersion to a negative binomial distribution. A total of 141 microarray gene expression datasets across 20 cancer types were downloaded from the PRECOG resource (Gentles et al., 2015). Normalized breast cancer gene expression and copy number alteration (CNA) datasets from METABRIC (n = 1,992) were downloaded from the European Genome-Phenome Archive (http://www.ebi.ac.uk/ega/) under the accession number EGAS00000000083. CRISPRi screening data on functional lncRNAs in MDA-MB-231 and K562 cell lines were downloaded from Liu et al. (2017).

Construction of cancer-specific regulons

The ARACNe-AP algorithm was applied to each processed TCGA RNA-seq cancer dataset using the TANRIC lncRNA Ensembl gene IDs as the regulator mapping set. Briefly, ARACNe-AP calculates the mutual information between a lncRNA and potential target genes and removes edges that are unlikely to represent a biological link using the concept of data processing inequality (Margolin et al., 2006; Lachmann et al., 2016). We implemented the algorithm using 100 bootstrap iterations and a p-value threshold of 0.01. Each regulon in the cancer-specific network consisted of a lncRNA and its associated genes. Each edge in the regulon was assigned a weight using the mutual information scores outputted by ARACNe-AP (Alvarez et al., 2016). The mutual information scores were divided by the maximum score within each regulon so that they had a range from 0 to 1. The sign of the edge was assigned by computing a Pearson correlation coefficient between the lncRNA’s expression and the associated gene’s expression across the samples. Since genes in a regulon are positively or negatively correlated with the corresponding lncRNA in a specific cancer type, their expression can be used to impute the expression level of the lncRNA.

Inference of lncRNA expression in microarray datasets

For each cancer-specific regulon, we defined a pair of profiles—the genes with a positive weight were assigned to an “up-regulated” profile and the genes with a negative weight were assigned to a “down-regulated” profile. In the up-regulated profile, all genes that had a negative weight were assigned a value of 0 and all genes in the down-regulated profile that had a positive weight were assigned a value of 0. The values in the down-regulated profile were then forced to be positive. Only profiles with 20 or more associated genes were used. Thus, each lncRNA was assigned two regulon weight profiles that capture the magnitude and direction of the genes it was associated with. Genes with higher weights in the two profiles will contribute more to the imputation of lncRNA expression.

After constructing the regulon weight profiles, lncRNA expression was inferred in microarray datasets by using the regulon weight profile derived from the same cancer type (or most related cancer type) as the microarray experiment (Supplemental Results). To apply the regulon weight profiles to infer lncRNA expression in microarray samples, we utilized the BASE algorithm (Cheng et al., 2007) which outputs a predicted expression value for each lncRNA in every patient sample. BASE imputes the relative expression level of a lncRNA based on the expression of genes that it correlates with (i.e., regulon genes). Specifically, the algorithm sorts each patient’s gene expression profile from highest to lowest expressed genes and weights them using the two regulon weight profiles. BASE then calculates a running sum statistic by moving down the profile and calculating a foreground function which captures the weighted enrichment of the lncRNA’s associated genes at the top and bottom of the patient’s gene expression profile. The foreground function is then compared to a background function and the maximum deviation between the foreground and background functions is computed. The maximum deviation calculated from the down-regulated profile is subtracted from the maximum deviation calculated from the up-regulated profile to yield a pre-inferred lncRNA expression value (pre-iExpr). For a lncRNA, if positively associated genes tend to be highly expressed (at the top of the expression profile) in a tumor sample while negatively associated genes tend to be lowly expressed (at the bottom of the expression profile), we will observe a high pre-iExpr value. The patient’s gene expression profile is then randomly permuted and the procedure is repeated; this is performed 1,000 times to yield a null pre-iExpr distribution. The pre-iExpr score is then normalized by dividing by the mean of the null pre-iExpr values to yield the final inferred expression of the lncRNA (iExpr). The formulas describing the details of this algorithm are provided in the Supplemental Methods.

Systematic inference of prognostic lncRNAs

A univariate Cox proportional hazards model was fit to the inferred and actual expression values for each lncRNA, separately for TCGA, PRECOG and METABRIC datasets. Actual expression was available for a small set of lncRNAs in the PRECOG and METABRIC datasets and were used for downstream validation. From the models, z-scores were calculated by dividing the Cox regression coefficient by its standard error. A z-score < 0 indicates that a lncRNA is protective and positively associated with survival. Conversely, a z-score > 0 indicates that a lncRNA is hazardous and negatively associated with survival.

In the PRECOG dataset, there were several microarray datasets belonging to the same cancer type. After computing the inferred expression for all lncRNAs within each dataset, we fitted a univariate Cox regression model to measure the association between a lncRNA and all-cause or disease-specific mortality (if available). z-scores were extracted from the fitted models and a meta z-score was calculated for each lncRNA across all the microarray datasets belonging to the same cancer type. The meta z-score was calculated using weighted Stouffer’s z-score method using the dataset sample size as weights. A meta z-score < 1 indicates a positive association and a meta z-score > 1 indicates a negative association with survival. In addition, robust meta z-scores were calculated for each lncRNA by leaving out the dataset yielding the most significant association and repeating the procedure. Meta p-values were calculated from the meta z-scores by referring to the standard normal distribution. Meta p-values were adjusted for each cancer type using Benjamini–Hochberg and Bonferroni multiple testing correction methods. Kaplan–Meier analysis of lncRNAs was performed by dichotomizing patients into high (>0) and low (<0) inferred lncRNA expression groups and performing a log-rank test to calculate statistical significance.

In the METABRIC dataset, a multiple Cox regression model was applied and included age at diagnosis, tumor size, stage, ER and HER2 status as covariates. Disease-specific mortality was used as the outcome.

Validation of survival analysis

To compare survival results across datasets, we performed two validation analyses: (1) Cross-dataset analysis comparing Cox regression results using actual lncRNA expression from TCGA with results using inferred lncRNA expression from PRECOG and (2) Within-dataset comparison of survival results generated by models fitted to inferred or actual lncRNA expression in PRECOG and METABRIC. Pearson correlation was used to evaluate the consistency between lncRNA regression z-scores derived from actual and inferred expression within and between datasets. A one-sided Fisher’s exact test was used to compute the enrichment of prognostic TCGA lncRNAs (actual lncRNA expression) in the set of prognostic PRECOG lncRNAs (inferred lncRNA expression). Prognostic lncRNAs were selected using FDR < 0.05 and non-prognostic lncRNAs were selected using a FDR > 0.1. Protective (hazard ratio < 1) and hazardous (hazard ratio > 1) lncRNAs were analyzed using separate enrichment tests.

Revealing lncRNA-based subtypes in breast cancer

In the METABRIC dataset, feature selection was performed by selecting the top 500 lncRNAs with the highest variation of inferred expression across patients. The inferred expression levels were then z-transformed across patients and gene-wise unsupervised clustering was performed using Euclidean distance and complete linkage.

Analysis of prognostic and essential lncRNAs

Hazardous lncRNAs identified from the PRECOG meta-analysis of breast cancer and hematopoietic cancer datasets were selected using a z-score cutoff of >0 and p-value cutoff of ≤0.1. LncRNA functional screening data were downloaded from Liu et al. (2017) and contained averaged phenotype scores derived from systematic CRISPRi knockout of lncRNAs. Essential lncRNAs were defined as those that when knocked down, result in ablation of cell proliferation and cell death and was quantified by a phenotype score included in the dataset. Essential lncRNAs in the MDA-MB-231 (breast) or K562 (hematopoietic) cell lines were selected using an average phenotype score cutoff of <0 and a p-value cutoff of ≤0.1. The average phenotype score measured the growth effect on the cell line when a particular lncRNA has been knocked down; a value <0 indicated essentiality and a value >0 indicated non-essentiality. The enrichment overlap between essential MDA-MB-251 lncRNAs and hazardous breast cancer lncRNAs was computed using a one-sided Fisher’s exact test. The same test was used to calculate the enrichment overlap between essential K562 lncRNAs and hazardous hematopoietic cancer lncRNAs.

Prognostic lncRNAs and CNAs

Long non-coding RNAs associated with prognosis in the METABRIC dataset were mapped to the genome for each patient. Hazardous lncRNAs were selected using z-score > 0 and FDR ≤ 0.01 as the cutoff. Protective lncRNAs were selected using z-score < 0 and FDR ≤ 0.01 as the cutoff. The CNA dataset provides the copy number signal of genomic segments throughout the genome for each patient along with binary calls indicating amplification (1) or deletion (−1). For each patient, a Fisher’s exact test was performed to measure significant enrichment of hazardous lncRNAs (compared to protective lncRNAs) in genomic segments that had undergone copy number amplification or deletion. When constructing the contingency table for a Fisher’s exact test, every cell had to have at least five counts in order for the test to be performed for the patient to ensure robust enrichment results. In total, 1,595 METABRIC patients were used to test enrichment of hazardous lncRNAs in amplified regions and 901 patients were used to test enrichment of protective lncRNAs in deleted regions. The Benjamini–Hochberg procedure was used to adjust for multiple hypothesis testing. When calculating the CNA signal corresponding to each lncRNA, the average copy number signal of all segments overlapping the gene region (transcription start site to the termination site) was used.

When performing the CNA enrichment analysis in the TCGA dataset, lncRNAs associated with prognosis in glioblastoma or ovarian cancer were selected using an unadjusted p-value cutoff of <0.05. An FDR cutoff of 0.1 was used to identify prognostic lncRNAs in pancreatic cancer and lung adenocarcinoma. These significance cutoffs were chosen to ensure a sufficient number of prognostic lncRNAs for enrichment analysis. Segments were selected using a CNA signal of >0 and <0 for amplification and deletion, respectively.

Results

Overview of analysis

To systematically identify lncRNAs associated with patient prognosis, we applied the ARACNe-AP algorithm (Lachmann et al., 2016) to 23 TCGA RNA-seq datasets to generate lncRNA regulons for each cancer type. ARACNe-AP calculates the mutual information between a lncRNA and potential target genes and removes edges that are unlikely to represent a biological link using the concept of data processing inequality (Margolin et al., 2006; Lachmann et al., 2016). The resulting regulons represent a network where the edges encode the magnitude and direction of association between lncRNAs and other genes based on their gene expression across samples (See “Methods”). A lncRNA’s expression can be inferred within a microarray dataset lacking lncRNA probes by analyzing the aggregate expression of the protein coding genes composing that lncRNA’s regulon. In total, we generated cancer-specific lncRNA regulons for 23 different cancer types using TCGA RNA-seq datasets. Once these regulons were generated, we transformed them into weight profiles and validated their predictive accuracy in TCGA. We then extended our analysis by inferring lncRNA expression in microarray data compendia from PRECOG and METABRIC using the regulon weight profiles and the BASE algorithm (Cheng et al., 2007). The BASE algorithm outputs a predicted expression value for each lncRNA in every patient sample by imputing the relative expression level of a lncRNA based on the expression of genes that it correlates with (i.e., regulon genes). Regulon weight profiles were selected to interrogate microarray data based on matched cancer type. After inferring the expression of thousands of lncRNAs, we performed a systematic pan-cancer screen for prognostic lncRNAs using survival information included in the microarray gene expression data compendia (Fig. 1).

Figure 1 Overview of analysis.

TCGA RNA-seq data from 23 cancer types were used as input into the ARACNe algorithm to generate cancer type specific lncRNA-target gene regulons. These regulons were used with the BASE algorithm to infer lncRNA expression in PRECOG and METABRIC microarray datasets. The BASE algorithm infers the expression of lncRNAs in microarray data using the aggregate expression of the lncRNAs’ associated protein coding genes. Lastly, a systematic pan-cancer analysis of 9,463 lncRNAs was carried out to identify prognostic lncRNAs across 20 different cancer types in the microarray data compendia.

Inferred lncRNA expression strongly correlates with actual expression

By implementing the ARACNe-AP algorithm in TCGA RNA-seq datasets, we constructed thousands of lncRNA regulons for each TCGA cancer type (Fig. 2A). Each regulon contains a lncRNA and its associated genes, which can be used as features to infer that specific lncRNA’s expression. An example of an inferred lncRNA expression pattern and the expression of its associated genes is provided in Fig. S1. The number of regulons varied across cancer types depending on whether any genes were found to have high mutual information with any given lncRNA based on expression signal. To confirm that the inferred expression of the lncRNAs was indeed accurate, we correlated each lncRNA’s inferred expression with its actual expression in all TCGA RNA-seq datasets. We observed that for the majority of lncRNAs, their inferred and actual expression were highly correlated across 23 cancer types as shown by the left-skewed distribution of correlation coefficients (Fig. 2B). These results indicate that it is possible to infer the expression levels of lncRNAs based on the aggregate expression of its associated genes.

Figure 2 Comparison of inferred lncRNA expression and actual lncRNA expression.

(A) Number of lncRNA regulons identified in 23 TCGA cancer types from the ARACne algorithm. Each regulon consists of a lncRNA and its associated protein coding genes. (B) Distribution of Spearman correlation coefficients from comparing inferred lncRNA expression with its actual expression using RNA-seq data from 23 TCGA cancer types.

Furthermore, we inferred lncRNA expression in the METABRIC dataset and compared the inferred and actual expression of 95 lncRNAs, which had probes present in the microarray platform. We observed that 82 of these lncRNAs had inferred expression values positively correlated with probe expression with 59 having significant associations (Fig. 3A; p ≤ 0.05). As an example, the correlation between the inferred and actual expression of HOTAIR and PVT1 was 0.54 and 0.60, respectively (Figs. 3B and 3C). This analysis was repeated in each PRECOG dataset and we again observed that the correlation coefficient distributions were left-skewed indicating that approximately 95% of the inferred lncRNA expression values were positively correlated with actual probe expression (Figs. 3D–3F), with a median correlation coefficient of 0.6. These results demonstrate that lncRNA expression can be inferred using the expression of protein coding genes in microarray datasets. Furthermore, we show our lncRNA inference platform is robust and can be generalized to different datasets as demonstrated by our analysis of TCGA, PRECOG and METABRIC.

Figure 3 Comparison of inferred and actual lncRNA expression in METABRIC using available probes.

(A) Waterfall plot showing correlation of inferred lncRNA expression and lncRNA probe expression in the METABRIC microarray dataset. Each lncRNA that had an available probe in the METABRIC microarray platform was selected to compare its inferred expression with its actual expression using Spearman correlation. Scatterplots show correlation of inferred and actual expression for (B) HOTAIR and (C) PVT1. (D–F) Distribution of correlation coefficients between inferred lncRNA expression and actual probe expression for 141 microarray datasets across 20 cancer types in the PRECOG compendium. Dashed vertical line indicates no correlation. Panels separated to increase legibility. (G) Heatmap showing inferred lncRNA expression differences between Luminal A, Luminal B, Normal-like, HER2-enriched and Basal breast cancer subtypes. Color bar shows z-score spectrum.

Previous studies have shown that the expression patterns of lncRNAs recapitulate the four well-known molecular subtypes in breast cancer (Su et al., 2014), which are associated with tumor behavior and patient prognosis (Perou et al., 2000). We sought to confirm whether inferred lncRNA expression could similarly distinguish between the different breast cancer subtypes. Using the METABRIC dataset, we performed hierarchical clustering of the genes using the inferred expression values of 500 lncRNAs with highest variance across samples and indeed found subtype-specific differences in inferred lncRNA expression (Fig. 3G). This finding implies that lncRNA activity varies across breast cancer molecular subtypes and may play a role in tumor behavior.

Exploring the prognostic landscape of lncRNAs across 20 tumor types

In light of the current dearth of RNA-seq datasets with survival metadata and the expansive trove of microarray datasets that do have this valuable clinical information, we first used the PRECOG compendia to systematically infer lncRNA expression. Datasets in PRECOG all include patient survival information, and many patients within these datasets have been followed-up for longer periods of time compared to those recorded in TCGA, offering greater statistical power when performing survival analyses (Clark et al., 2003). We carried out a systematic inference of prognostic lncRNAs in PRECOG datasets with a sufficient number of probes (500) and performed a meta-analysis by combining the results within each tissue type. From this systematic screen, we identified a number of prognostic lncRNAs associated with both increased and decreased patient mortality risk in 13 PRECOG cancer types (Fig. 4A). These results suggest that lncRNAs may play a substantial role in the progression of all neoplasia. An overview of all lncRNAs and their associations with prognosis is available in the Supplemental Results.

Figure 4 Systematic screening of prognostic lncRNAs in PRECOG compendium.

(A) Table showing the number of lncRNAs identified to be associated with patient prognosis across PRECOG cancer types using standard and robust meta-analysis methods. Different cutoffs for the Benjamini meta-FDR and Robust Benjamini-FDR are displayed. Haz. indicates hazardous and Pro. indicates protective. (B) Selected prognostic lncRNAs (Adjusted p < 0.001) and their association with prognosis in each cancer type. White cells indicate associations with p > 0.05 or lncRNAs whose expression cannot be inferred in that cancer type. Bar plots showing odds ratio indicating enrichment overlap between essential lncRNAs in (C) K562 cells and (D) lncRNAs associated with prognosis in hematopoietic cancers in PRECOG. Bar plots showing odds ratio indicating enrichment overlap between essential lncRNAs in (E) MDA-MB-231 cells and (F) lncRNAs associated with prognosis in breast cancer in PRECOG. Haz. indicates hazardous and Pro. indicates protective. Kaplan–Meier plots showing association of (G) TINCR; (H) H19; (I) EGOT and (J) RP11-108P20.4 inferred expression with patient prognosis in selected brain, liver, breast cancer and prostate cancer datasets from PRECOG.

Furthermore, we found that some lncRNAs including HOTAIR and H19 were associated with poor survival across multiple cancer types (Fig. 4A). These results are in accordance with several reports implicating these lncRNAs in neoplastic progression and metastasis across different cancer types (Matouk et al., 2007; Gupta et al., 2010). It has been suggested that HOTAIR promotes cancer invasiveness and metastasis by the induction of a more embryonic-like state (Gupta et al., 2010), which leads to increased resistance to known therapies and is a marker for poor prognosis in almost all cancer types (Ge et al., 2017; Shibue & Weinberg, 2017). However, some lncRNAs like TINCR were associated with both good and poor prognosis depending on the cancer type (Fig. 4B). Particularly, TINCR was associated with unfavorable prognosis in breast cancer, which is consistent with the implication of TINCR in promoting breast cancer tumorigenesis (Xu et al., 2017; Liu et al., 2018). TINCR can stabilize mRNA by preventing Staufen-mediated mRNA decay of differentiation genes in epidermal tissue (Kretz et al., 2013), but it is unclear whether this mechanism plays a role in tumor evolution.

To provide evidence that the identified prognostic lncRNAs are functionally relevant, we performed an integrative analysis of CRISPRi data generated from a high-throughput, systematic screen for lncRNAs that are essential for cancer cell growth (Liu et al., 2017). We calculated the enrichment of functional lncRNAs identified in MDA-MB-231 and K562 cell lines in the set of prognostic lncRNAs identified in breast and hematopoietic cancers, respectively. We observed that hazardous lncRNAs in hematopoietic cancers were enriched in essential lncRNAs (Odds ratio = 2.26, p = 7.1E−5) and protective lncRNAs were depleted in essential lncRNAs (Odds ratio = 0.73, p = 0.16) indicating that hazardous lncRNAs in hematopoietic cancers tend to be required for cancer cell growth compared to protective or non-prognostic lncRNAs (Figs. 4C and 4D). Likewise, we discovered that hazardous lncRNAs in breast cancer was also enriched in essential lncRNAs (Odds ratio = 3.04, p = 5.4E−3) and protective lncRNAs were depleted in essential lncRNAs (Odds ratio = 0.30, p = 3.2E−4), again suggesting that hazardous lncRNAs are more likely to be essential because they contribute to cell growth and are thus associated with increased mortality risk (Figs. 4E and 4F). Together, these results indicate that hazardous lncRNAs identified in our analysis of breast and hematopoietic cancers are functionally relevant, at least in the context of in vitro cancer cell growth.

In addition to known cancer-associated lncRNAs, our screen also generated novel hypotheses about lncRNAs that have not been well-studied in certain cancer types. To highlight, high TINCR inferred expression was associated with improved survival in patients with brain tumors (Fig. 4G; p = 5E−8). Moreover, high H19 inferred expression was associated with decreased mortality risk among patients with liver tumors (Fig. 4H; p = 0.002). EGOT inferred expression was associated with decreased mortality risk in patients with breast cancer (Fig. 4I; p = 9E−11). This is consistent with a previous study, which showed that downregulation of EGOT correlates with worse clinicopathological features and poor prognosis in breast cancer (Xu et al., 2015). Lastly, we found that high inferred RP11-108P20.4 expression was associated with improved survival in prostate cancer (Fig. 4J; p = 2E−15), which coincides with a recent report introducing RP11-108P20.4 as part of a four lncRNA gene prognostic risk signature for prostate cancer (Huang et al., 2002). These results demonstrate that novel prognostic lncRNAs can be identified across several cancer types from common microarray datasets.

Furthermore, to assess the reproducibility of our screen, we performed a survival analysis of 23 TCGA cancer types to identify prognostic lncRNAs using their actual expression in each dataset. From this screen, prognostic lncRNAs (FDR < 0.05) were identified in five cancer types (LUAD, LGG, BLCA, LIHC and LAML). We stratified the lncRNAs into protective or hazardous and computed their enrichment, respectively, in protective or hazardous lncRNAs predicted from the PRECOG datasets. We identified significant overlap between the two sets of prognostic lncRNAs in all five cancer types (Fig. 5A). Moreover, we compared the Cox regression z-scores (TCGA) and meta z-scores (PRECOG) for all lncRNAs within lung adenocarcinoma, low-grade glioma and bladder cancer datasets and observed significant correlations (Figs. 5B–5D). These z-scores were calculated by dividing the Cox regression coefficient by its standard error and the meta z-scores were calculated using weighted Stouffer’s z-score method using the dataset sample size as weights. Several well-studied lncRNAs including H19, BCAR4, GAS5, XIST, HOTAIR and EGOT had concordant z-scores (Figs. 5B–5D).

Figure 5 Prognostic lncRNAs identified in TCGA and PRECOG.

(A) Barplots showing odds ratios from enrichment analysis of prognostic lncRNAs identified in TCGA and PRECOG for LUAD, LGG, BLCA, LIHC and LAML. Enrichment analysis was performed separately for lncRNAs with hazard ratios >1 (Red) and <1 (Blue). Vertical black line denotes an odds ratio of 1. Scatterplots showing correlation of z-scores and meta z-scores for all lncRNAs screened in TCGA and PRECOG, respectively, in (B) lung cancer, (C) brain cancer and (D) bladder cancer. Labeled points denote lncRNAs that have been characterized in previous literature.

lncRNAs associated with prognosis localize to genomic regions under selective pressure

Operating under the hypothesis that genomic amplifications and deletions indicate regions of positive and negative selective pressure by the tumor, respectively (Zack et al., 2013), we aimed to provide further evidence that lncRNAs associated with prognosis are also linked to genomic structural abnormalities that confer a selective advantage to neoplastic cells. Thus, we analyzed CNA together with inferred lncRNA expression of each patient sample in the METABRIC data set. Strikingly, we observed a significant enrichment of lncRNAs associated with poor prognosis (hazardous) in amplified regions of the genome in 448 METABRIC patient tumors (Fig. 6A). In comparison, we only observed 54 patient tumors where amplified regions were significantly depleted of hazardous lncRNAs (Fig. 6B). Likewise, we observed 47 patients with deleted genomic regions enriched in lncRNAs associated with decreased mortality (protective), compared to 17 patients who had protective lncRNAs depleted in deleted genomic regions. We also explored whether prognostic lncRNAs were enriched in amplified or deleted regions of the genome in pancreatic cancer, lung adenocarcinoma and glioblastoma TCGA datasets and observed consistent results (Fig. S2). In summary, these results indicate that prognostic lncRNAs localize to genomic regions that undergo CNA suggesting that they are under both positive and negative selective pressure by the tumor.

Figure 6 Enrichment of prognostic lncRNAs in genomic regions with copy number alterations.

Waterfall plots showing enrichment of (A) hazardous and (B) protective lncRNAs in amplified and deleted regions of the genome for each patient, respectively. Log2 odds ratio >0 indicates enrichment and log2 odds ratio <0 indicates depletion. Purple bars indicate statistically significant enrichment. (C) High JRK inferred expression is associated with poor prognosis. (D) High JRK inferred expression is concentrated in regions with higher copy number signal across all METABRIC patients. (E) High CADM3-AS1 inferred expression is associated with favorable prognosis. (F) High CADM3-AS1 inferred expression is concentrated in genomic regions with lower copy number signal across all METABRIC patients.

To demonstrate that lncRNAs associated with mortality risk are under selection, we highlight JRK and CADM3-AS1. In our analysis of JRK, we discovered that patients with high inferred JRK expression exhibited a higher mortality rate compared to patients with low inferred JRK expression (Fig. 6C). In conjunction with this result, we also observed higher amplification signal of the region harboring JRK in patients with high inferred JRK expression compared to patients with low inferred JRK expression (Fig. 6D). These results suggest that JRK exhibits a pro-oncogenic effect because it is under positive selection by breast tumors, which consistently coincides with their association with increased mortality risk. In contrast, we found that high CADM3-AS1 inferred expression was associated with a more favorable prognosis compared to patients with low CADM3-AS1 inferred expression (Fig. 6E). In agreement with our prediction, we found that the genomic region harboring CADM3-AS1 was significantly more amplified in patients with low CADM3-AS1 expression compared to patients with high CADM3-AS1 expression (Fig. 6F). Together, these results suggest that due to CADM3-AS1’s association with decreased mortality risk, it exhibits anti-tumor effects that are not selected for by breast neoplasms. Our analysis of prognostic lncRNAs in the context of CNA indicates that they are under selective pressure and provides evidence that they are functionally involved in cancer development. This hypothesis compliments a recent publication showing that somatic copy number variations in lncRNA loci were predictive of target gene expression and might be responsible for the dysregulation of dozens of cancer-associated genes (Chiu et al., 2018).

Discussion

Investigation into lncRNAs using integrative and systematic approaches can help provide insight into the genome’s “dark matter” and how it may influence disease and ultimately patient prognosis in cancer. Studies are now underway to characterize and dissect the intricacies of lncRNA regulatory mechanisms in several biological contexts which may revise our current understanding of genome regulation (Cech & Steitz, 2014). Hypothesis generating projects are essential for guiding the biomedical community towards investigating more promising leads as to accelerate the discovery of novel drug targets and biomarkers for cancer and other diseases. We have proposed a novel analysis framework to infer lncRNA expression in microarray gene expression data compendia and subsequently carry out systematic survival analysis to identify prognostic lncRNAs across 20 cancer types. Our approach is novel in that we utilize expression information from TCGA RNA-seq data to generate cancer-specific lncRNA regulon profiles that capture the lncRNA-gene relationships within a specific tissue context. We then apply these profiles to microarray gene expression data using a sensitive enrichment algorithm, BASE, to infer lncRNA expression based primarily on protein coding gene expression. We perform this analysis at a pan-cancer scale to identify new prognostic lncRNAs that have global and tissue-specific associations with survival. In contrast to other prognostic lncRNA pan-cancer analyses, we evaluated lncRNA expression in a large number of microarray datasets, providing us with a more comprehensive view of prognostic lncRNAs in cancer.

In particular, we were able to validate that the inferred lncRNA expression values are accurate and reproducible within and across several datasets. We identified novel associations between lncRNAs and patient mortality risk across 13 (out of 20 total) cancer types that can be further evaluated in more detail. We confirmed that associations between lncRNA expression and mortality risk were consistent regardless of whether inferred or actual expression were used and showed that prognostic lncRNAs identified in breast and hematopoietic cancers were significantly enriched in functional lncRNAs required for cell growth. Furthermore, we demonstrated that hazardous lncRNAs were enriched within regions under positive selective pressure.

In spite of the evidence we provide, this study does have limitations that are imposed by the data. First, in each microarray dataset, we used the regulon profile that was generated from the TCGA cancer type that was the best match based on tissue. However, it was not always possible to find an exact cancer type match for each microarray dataset. Thus, using an inappropriately matched regulon profile may yield false associations. Second, univariate Cox regression models were used to screen for prognostic lncRNAs, which do not account for other clinical or demographic factors that may modify the associations. These may include age, gender, race, histological marker status, stage and grade. However, as an initial screen our framework can be further improved to include multivariate analyses in follow up studies if more specific hypotheses are to be tested (McNamee, 2005). Third, not all lncRNAs are poly-adenylated and are thus captured in poly-(A)-enriched RNA-seq or microarray studies. Due to this, we likely did not include all known lncRNAs in our study. Lastly, our analysis does not account for all cancer subtypes to address the issue of molecular heterogeneity within the same cancer type (Gerdes et al., 2014). As a result, certain associations between lncRNA expression and prognosis may only be valid in certain subtypes of the same cancer type. As stated previously, future analyses can address this issue by performing subgroup analyses within specific subtypes.

Conclusions

Our approach can be extended to other microarray gene expression datasets by utilizing our regulon profiles to infer lncRNA expression. As a result, it is possible to identify novel associations between lncRNAs and other disease phenotypes other than survival. Moreover, is possible to generate regulon profiles for other non-coding RNA species and infer their expression in microarray datasets. In summary, our systematic analysis introduces new avenues to investigate clinically relevant lncRNAs and demonstrate that these long, diverse transcripts constitute a new source of gene products that can serve as novel drug targets or biomarkers.

Supplemental Information

Supplemental Information 1 Example of inferred lncRNA expression and expression of 20 most influential genes in the lncRNA regulon.

The expression of twenty genes that have the highest contribution to the inferred expression (iExpr) of lncRNA LINC01684 are depicted. The entire regulon consisted of 820 genes.

Click here for additional data file.

Supplemental Information 2 Enrichment of prognostic lncRNAs in regions with copy number alterations in TCGA.

(A) Enrichment of hazardous and protective lncRNAs in amplified and deleted regions of the genome, respectively, across lung adenocarcinoma patients. (B) Enrichment of hazardous and protective lncRNAs in amplified and deleted regions of the genome, respectively, across pancreatic cancer patients. (C) Enrichment of hazardous and protective lncRNAs in amplified and deleted regions of the genome, respectively, across glioblastoma patients.

Click here for additional data file.

Supplemental Information 3 Formulas describing the details of a number of methods used in this paper.

Click here for additional data file.

Supplemental Information 4 All significant associations in all evaluated microarray datasets between inferred lncRNA expression and overall survival.

Click here for additional data file.

Additional Information and Declarations

Competing Interests

Author Contributions

Data Availability

The authors declare that they have no competing interests.

Matthew Ung conceived and designed the experiments, analyzed the data, prepared figures and/or tables, authored or reviewed drafts of the paper, and approved the final draft.

Evelien Schaafsma analyzed the data, prepared figures and/or tables, authored or reviewed drafts of the paper, and approved the final draft.

Daniel Mattox analyzed the data, prepared figures and/or tables, and approved the final draft.

George L. Wang analyzed the data, prepared figures and/or tables, and approved the final draft.

Chao Cheng conceived and designed the experiments, analyzed the data, prepared figures and/or tables, authored or reviewed drafts of the paper, and approved the final draft.

The following information was supplied regarding data availability:

All significant associations between inferred lncRNA expression and patient prognosis are available as a Supplemental File.

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
