# Peer review of "Pan-cancer systematic identification of lncRNAs associated with cancer prognosis"

_PeerJ, doi:10.7717/peerj.8797_

## Round 0.1 · original submission · Minor Revisions

The reviewers have provided helpful and detailed comments on your manuscript. I feel that each of these comments have merit, so I ask that you address each of them. In particular, unless you can provide strong justification for including XIST in your validation descriptions (for example, if you are just focusing on females), then you should use a different gene.

Overall, the figures are well done. However, some of them are unsuitable to be interpreted by people who are colorblind (see https://www.somersault1824.com/tips-for-designing-scientific-figures-for-color-blind-readers/). Figures 1, 4, 5, and 6 are probably fine. But Figures 2 and 3 would benefit from some adjustments. However, I'm not sure exactly what to recommend. In Figure 2B, the bars are colored based on cancer type, but the bars are so narrow that even a color-seeing person can not distinguish between them. Please look at other ways to visualize this information. I have a similar comment for Figure 3B. On Figure 3C, colors are used for each of the cancer subtypes, but a colorblind friend palette would be better.

Addressing these comments will make your manuscript more accessible to those who read it.

Reviewer 1 ·

Basic reporting

reporting is generally good, some clarifications are needed.

Experimental design

the experimental design is sound and novel as it uses an ingenious way to use previously published microarray data.

Validity of the findings

the findings are interesting, albeit some clarifications are needed.

Additional comments

Review

The authors described a novel computational framework, in which they first applied the ARACNe framework to the TCGA pan-cancer RNA-seq data and calculated an essentially protein-coding-gene / lncRNA co-expression network based on mutual information theory (regulons). Next, they fit these regulons to the older microarray expression datasets and inferred the expression of lncRNAs in these microarray studies, which were not measure originally. By doing so, the authors were able to infer the lncRNA expression data for many of these older microarray-based studies, which also had longitudinal expression and clinical data for the patients.

This allowed the authors to conduct survival analysis on these lncRNAs and they identified a number of prognostic lncRNAs from these datasets. These lncRNAs were further grouped into “hazardous” and “protective”, according to the result of survival analysis. The authors then compared these lncRNAs with results from CRISPRi screen and found hazardous lncRNAs are enriched in “essential lncRNAs”.

Overall the work is quite interesting and the results are largely reasonable. I have the following comments that the authors need to address before the paper can be further considered for publication.

Figure 2B is shown as evidence that the regulon-based inference approach works in inferring the lncRNA expression level. It appears to me that the signal is stronger for certain cancer type than others. I would like to ask the authors to show the distributions individually for each cancer type, and comment on why the difference in the inference performance. Also, correlations are not the best indicator of accuracy, the correlations should be compared with random controls, i.e. what is the correlation between random pairs of predicted values and real measured values?



Figure 3A.
How is the p-value calculated (which is shown as a red horizontal bar) ? The red bar happens to intersect with the Y axis at about 0.05 too, is this a coincidence or did the authors actually meant Pearson rho cutoff at 0.05 ? The correlation for HOTAIR and XIST, 0.54 and 0.53. Are these high ? low ? compared to what ? I think the authors need to provide some perspective here.
Also, XIST is not the most appropriate lncRNA for such comparison since it is X-linked. Is the comparison only limited to female subjects ?

Figure 3B.
Again, if you take random pairs of predicted and experimentally measured lncRNAs and calculate Spearman correlation, are the correlations centered at 0.0 ?

Reviewer 2 ·

Basic reporting

Authors should mention that lncRNAs analyzed in the TCGA and microarray datasets are polyadenylated, and that there are many other lncRNAs which are non-polyadenylated, and therefore, not analyzed in the present study. Indeed, majority of lncRNAs are non-polyadenylated.

Experimental design

Authors should perform general descriptions in the Results sections about:

- ARACNe-AP algorithm
- BASE algorithm
- Cox regression z-scores
- Meta z-scores

Althought a deep description of this methods are explained in the M&M sections, brief explanations are necessary in the Results.

Validity of the findings

no comment

Additional comments

Manuscript describes a profound bioinformatic analysis in order to extract lncRNAs associated with poor outcome in different cancer types. I would like to see whether the lncRNAs with prognostic association are mapping into regulatory regions within the human genome. More specifically, how many of them are within enhancer regions? It would also be interesting to see whether the coding genes within lncRNA-regulons are located close to the corresponding lncRNA in the genome, so they could represent enhancer-promoter regulations, or sense (coding) to anti-sense (non-coding) interactions.

---

## Round 0.2 · Minor Revisions

Thank you for addressing the reviewers' comments. There is just one minor change that needs to be made. The latest Figure 3C uses a color palette that includes both red and green shades. However, the most common form of color blindness is red-green colorblindness. Perhaps you can justify this, but my understanding is that the palette you are using here will be difficult for people with red-green colorblindness to decipher. Please update that and resubmit. Perhaps helpful would be this tool: http://colorbrewer2.org

---

## Round 0.3 · accepted · Accept

Thank you for adjusting the color palette in that figure.